# Body Image, Autonomy, and Vaccine Hesitancy: A Psychodynamic Approach to Anti-Vaccine Individuals’ Resistance

**DOI:** 10.3390/bs15040493

**Published:** 2025-04-08

**Authors:** Alberto Zatti, Nicoletta Riva

**Affiliations:** Department of Human and Social Sciences, University of Bergamo, 24129 Bergamo, Italy

**Keywords:** psycho-social attitude, psychodynamic defenses, body schema, social beliefs

## Abstract

This study examines the psychological and psychodynamic factors influencing vaccine hesitancy, focusing on body image and emotional processing. A cross-sectional observational design was used. Participants from five European countries completed the Body Image and Schema Test (BIST). ANOVA analyses compared cognitive, affective, and behavioral traits between pro- and anti-vaccine individuals. Findings indicate that anti-vaccine individuals exhibit higher levels of autonomy, distrust of authority, and emotional intensity, particularly in the form of heightened fear and anger. Their resistance to vaccination is linked to concerns about bodily integrity and a strong sense of self-protection, reflecting deep-seated psychological dispositions. This study highlights the role of defense mechanisms, personality traits, and social influences in shaping vaccine attitudes. By understanding these psychodynamic underpinnings, public health strategies can be better tailored to address vaccine resistance through targeted communication and interventions. The findings provide valuable insights for policymakers and healthcare professionals in designing more effective public health campaigns. The repository Open Science Framework link contains data, a complete presentation of the BIST theoretical framework, and a full description of the meaning of BIST Factors and Items.

## 1. Introduction

The 2020 pandemic lockdown led to months of social distance, eliciting diverse reactions across the population ([6]). Despite the rollout of the first COVID-19 vaccines in early 2021, a significant portion of the Italian and European population remained resistant to vaccination by September of the same year. Surveys on vaccine hesitancy conducted across various countries ([5]; [13]; [24]; [7]) revealed persistent skepticism, with health authorities implementing incentive policies, such as the Green Pass in Italy and Europe or vaccine lotteries ([15]), yet failing to engage substantial segments of the population.

As of October 2021, 9,829,232 eligible Italians, representing approximately 16% of the adult population, had yet to be vaccinated ([17]). Similarly, in France, approximately 12% of the adult population remained unvaccinated in the same period, with 4,820,000 individuals not receiving any doses. The unvaccinated adult population in Germany accounted for about 10%, equating to 6,700,000 individuals. The United Kingdom reported a slightly lower percentage, with 8% or 4,200,000 adults not pursuing vaccination. In Poland, the figures were higher, with around 20% or 6,000,000 adults not vaccinated. These Eurostat statistics provide a backdrop for this study, highlighting the varying levels of vaccine hesitancy across these countries and underscoring the importance of understanding the psychological and cultural factors that influence vaccine acceptance and resistance.

A growing body of research has sought to identify the psychological underpinnings of vaccine hesitancy ([28]; [29]; [27]). Studies have examined correlations with personality traits ([20]) and intellectual orientations ([16]), highlighting psychological factors that, while informative, remain inconclusive (see also [25]).

This study seeks to describe and interpret attitudes toward COVID-19 vaccination through the lens of Allport’s trait theory, which underscores the enduring personal characteristics influencing behavior. The central hypothesis posits that bodily experience—specifically the “affective evaluation” of COVID-19 vaccines—is pivotal in vaccine refusal. As [2] ([2]) outlined, emotional dispositions represent fundamental determinants of how individuals perceive and react to social phenomena, functioning as guiding forces behind conscious decision-making.

Vaccine acceptance and hesitancy have become pivotal topics in public health discourse, particularly after global vaccination campaigns. Understanding the psychological foundations behind these attitudes is crucial for developing targeted interventions.

Recent contributions from [9] ([9]) explore the pivotal impact of pre-existing attitudes on vaccine decisions, reinforcing the complexity of psychosocial interactions. The dynamics of vaccine-related information dissemination, as discussed by [14] ([14]), and [22] ([22]), provide a critical examination of how narratives shape individual choices. Furthermore, [26] ([26]) elucidate the substantial influence of social media on public health behaviors, illustrating a new vector for psychosocial research. The level of mistrust in public health systems, explored by [20] ([20]), underscores a deep-seated skepticism that must be addressed to mitigate resistance to vaccination efforts.

This study explores the psychodynamic and social-psychological characteristics of individuals who support vaccines (pro-vaccine) and those opposed to vaccines (anti-vaccine). The primary hypothesis posits that anti-vaccine individuals exhibit an anti-authoritarian personality structure, are more individualistic, experience invasion anxieties, and display pronounced anxiety and anguish. The Body Image and Schema Test (BIST) examines these traits through somato-psychic constructs, providing insights into body schema and emotional processing.

## 2. Theoretical Frameworks and Research Hypothesis

This study integrates social psychology concepts, such as conformity, authority resistance, and group dynamics, with psychodynamic theories focusing on unconscious drives, defense mechanisms, and body schema. The literature on psychological reactance ([11]) and anti-authoritarianism is explored alongside body image theory and emotional regulation models. Additionally, psychodynamic interpretations of somatic experiences are examined, highlighting the interplay between physical self-representation and emotional states ([12]).

In line with Allport’s concept of functional autonomy, vaccine hesitancy may be viewed as an attitude shaped by prior experiences and reinforced by current motivational states. The automatic, pre-reflective judgments described by [21] ([21]) can be reinterpreted within Allport’s framework as habitual predispositions rooted in an individual’s personal construct system ([18]). From this perspective, vaccine refusal may represent an attempt to preserve bodily integrity, consistent with [3]’s ([3]) notion of the “skin ego”, which serves as a psychological boundary against perceived threats.

While personality traits and intellectual orientations provide valuable insights, the complexities of vaccine hesitancy necessitate a broader exploration that encompasses body schema. This dimension, integral to understanding vaccine hesitancy, reveals how physical self-perception and emotional states significantly influence personal health decisions. Body schema does not merely respond to static traits but interacts dynamically with environmental cues and internal emotional states, thus shaping attitudes towards health interventions like vaccination.

Individuals with deeply ingrained personality traits, such as high levels of personal autonomy and skepticism toward authority, may exhibit a heightened resistance to vaccination. The conceptualization of attitudes as a triadic construct (Affect, Behavior, Cognition—ABC Model; [1]; [10]) provides a theoretical foundation for understanding the interplay of emotional reactions, behavioral tendencies, and cognitive beliefs surrounding vaccine decisions.

Recent research further supports Allport’s concept of the continuity of personality traits, illustrating that vaccine hesitancy is not an isolated attitude but a comprehensive stance influenced by historical and present psychological dynamics. For example, a study by [9] ([9]) highlights how pre-existing experiences with healthcare and authority figures can predispose individuals to resist vaccination, suggesting that vaccine hesitancy reflects enduring personality patterns and a reaction to current societal pressures. Similarly, [30] ([30]) underscores the role of social media and cultural factors in shaping these attitudes, confirming that vaccine hesitancy is deeply embedded in one’s continuous personal and psychosocial history.

These findings enrich our understanding by linking the concept of body schema with psychological and social realities. They affirm Allport’s theory that attitudes are shaped by prior experiences and reinforced by current motivational states but also that these “*memories*” are somehow embodied. This perspective deepens our comprehension of the psychological underpinnings of vaccine hesitancy and enhances the strategies for addressing it in public health initiatives.

[11] ([11]) suggests that perceived threats to personal freedom can lead to oppositional behaviors. Classical studies in conformity and authority, such as those by [4] ([4]) and [23] ([23]), examined how individuals respond to authority and group pressures. The dynamics between obedience, conformity, and resistance to authority figures show how people maintain self-decisions despite high social pressure. [31] ([31]) highlighted cultural and personality-based tendencies towards self-reliance versus group orientation.

Solomon [4]’s ([4]), Jack [11]’s ([11]), and Stanley [23]’s ([23]) foundational works are instrumental in this analysis, offering a lens through which we can view the current pandemic response. Their studies help to elucidate the underlying psychological mechanisms that may drive vaccine hesitancy, such as conformity, authority resistance, and the struggle for autonomy.

Insights into how high anxiety leads individuals to employ strategies to manage internal conflicts and external stressors unconsciously are particularly relevant ([8]; [32]). These mechanisms are crucial for understanding the complexities of vaccine hesitancy, where body autonomy and control are central themes. Similarly, the exploration of body schema contributes significantly to our understanding of how bodily perceptions influence psychosocial interactions.

[8]’s ([8]) research on the role of unconscious emotion in shaping behavior provides a robust contextual background. This integration bridges century-old psychological theories with contemporary methods, enriching our theoretical framework and enhancing the interpretation of our data.

The Internal Body Model ([34], BIST Manual) integrates bodily schema structure and sub-conscious self-image. In other words, the Body Image and Schema Test (BIST) describes how one evaluates self-individuals’ perceptions of their physical appearance and subconscious body awareness. It comprehends the emotional, cognitive, and functional dimensions affected by personal and cultural factors.

The Body Image and Schema Test (BIST) is a projective tool. It assesses individual perceptions and attitudes towards one’s body through various psychodynamic dimensions. This test includes multiple factors and items that evaluate the emotional, cognitive, and functional aspects of body schema and image. Factors such as the ‘Internal Body Model’, ‘Drive Inner Energy’, and ‘Ego Skin’ ([3]) help us understand how individuals perceive their internal sensations, motivational energies, and protective barriers against external threats. Each item within these factors aims to elicit responses that reveal deeper unconscious and conscious attitudes towards bodily integrity and autonomy.

It is important to note that the BIST is currently undergoing a rigorous validation process. This ongoing validation aims to ensure the reliability and accuracy of the test in measuring the constructs it purports to assess. By meticulously refining the BIST, researchers strive to enhance its applicability and precision in psychological research, particularly in studies exploring complex phenomena such as vaccine hesitancy.

The repository Open Science Framework link is available for a complete presentation of the BIST theoretical framework and a full description of the meaning of BIST Factors and Items: https://osf.io/vum2h/?view_only=216f282682ae409882d4c0d1439f8305 (accessed on 1 February 2025).

Through this lens, this study hypothesizes the following:

**H1:** 
*Individuals opposing vaccination will score higher in personality dimensions related to bodily integrity and self-assertion, particularly in traits associated with heightened vigilance and protective emotional responses.*


**H2:** 
*Anti-vaccine individuals will display elevated levels of emotional intensity, specifically heightened fear and anger, which may be traced to enduring personality patterns related to distrust and perceived control.*


**H3:** 
*Anti-vaccine individuals will demonstrate stronger adherence to individualistic values and a resistance to external control, reflecting a deeply rooted psychological disposition shaped by personal experiences and social context.*


In conclusion, Allport’s emphasis on the continuity of personality traits provides a valuable perspective for understanding the persistence of vaccine hesitancy. This study contributes to a nuanced understanding of public health challenges and the diverse psychological responses to vaccination efforts by examining the enduring personal characteristics that shape attitudes and behaviors.

## 3. Methods

### 3.1. Participants/Sample

Participants were recruited from Italy, France, the United Kingdom, Germany, and Poland. They were categorized by age (youth, adults, seniors) and stance on vaccines (pro-vaccine and anti-vaccine). Table 1 reports the numbers and distribution of the sample.

Data for this study were collected across five different European countries using the Qualtrics (Silver Lake) paid opt-in panel system, ensuring a diverse and representative population sample. The motivation for a cross-country approach was to test the hypothesis across different national contexts, thus broadening the applicability and relevance of the findings.

While the original segmentation of the sample included age and gender categories, these variables were not used in the statistical analysis presented in this paper due to space limitations. This decision was made to streamline the presentation and focus on the primary psychodynamic variables of interest.

Regarding the statistical assumptions, the analysis confirmed the homogeneity of variance across groups, normality in the distribution of responses, and independence of observations. These conditions ensure the validity of the reported ANOVA results, supporting the findings’ reliability in illustrating how body schema perceptions influence vaccine hesitancy across different demographic and national contexts.

Participants completed the Body Image and Schema Test (BIST), which evaluates individual perceptions of their body schema and emotional responses. A significant limitation of this study is the reliance on self-report measures to capture complex embodied emotional states. While these instruments provide valuable insights into personal attitudes and perceived experiences, they may not fully encapsulate the depth and variability of unconscious emotions and biases. This limitation is particularly relevant in vaccine hesitancy, where deep-seated anxiety and societal influences might not be fully articulated through self-report. Future research should consider integrating more objective measures, such as physiological responses, to complement self-reported data and provide a more comprehensive picture of the underlying emotional dynamics.

Although missing data were minimal, responses with incomplete data on primary psychometric measures were excluded from the final analysis to ensure the robustness of ANOVA comparisons. The reported final sample size reflects only complete cases.

Participants were recruited through an online opt-in panel and completed the survey electronically. All respondents who completed the questionnaire were included in the analysis; thus, no dropouts occurred, and a flow diagram was not deemed necessary.

### 3.2. Procedure

Participants completed an ad hoc questionnaire with statements about trust in authority and the degree of fear, anger, and anguish. The form included BIST, a projective assessment comprising 25 image-based items rated on a four-point scale. The test evaluates five primary psychological factors: Internal Body Model, Body Image, Psychomotor Function, Ego Skin, and Self-Image.

### 3.3. Statistical Analysis

Statistical analyses were conducted using ANOVA (Analysis of Variance) to compare mean scores between pro-vaccine and anti-vaccine groups across BIST factors.

Table 2 and Table 3 present statistically significant mean differences between pro-vaccine and anti-vaccine subjects in personal declarations and in the Body Image and Schema Test.

## 4. Results

### 4.1. Beliefs and Emotions Declared

As shown in Table 2, several items show highly significant differences (*p* < 0.001), indicating substantial group differences in attitudes towards authority and autonomy between pro-vaccine and anti-vaccine individuals. Anti-vaccine individuals exhibit a significantly stronger belief in personal autonomy and resistance to external control compared to pro-vaccine individuals. There is a marked tendency among anti-vaccine individuals to reject external impositions on their value systems, showing strong individualistic tendencies. Anti-vaccine individuals significantly prefer self-determination in health decisions, rejecting collective approaches to health. These results confirm a prevailing psychological profile of high autonomy and opposition to external control among anti-vaccine individuals.

Significant differences are observed in trust towards institutions and sources of information. Anti-vaccine individuals exhibit significantly lower trust in governmental institutions, suggesting skepticism towards health policies. A marked distrust in official information sources is evident, reinforcing their reliance on alternative sources of information. Anti-vaccine individuals have higher trust in information from social networks or peer groups, which may reinforce their beliefs and skepticism toward scientific data. These findings highlight a polarized trust pattern, where anti-vaccine individuals reject mainstream institutions but place greater trust in peer-driven information sources.

Items measuring fear and anxiety responses to perceived threats show significant differences between the two groups. Anti-vaccine individuals experience heightened anxiety regarding loss of control over their health decisions, reflecting a fear of external influence. The significant result indicates a strong fear of bodily invasion, which aligns with vaccine hesitancy due to perceived threats to bodily autonomy. Anti-vaccine individuals express anxiety over societal pressure to conform to majority health behaviors. These results emphasize that fear of control and bodily invasion play a crucial role in vaccine hesitancy.

Emotional responses related to anger in anti-vaccine individuals show a higher tendency of these individuals to express anger when their autonomy or goals are restricted. A strong rejection of imposed limits is evident, reinforcing their resistance to mandatory vaccination policies. Anti-vaccine individuals experience frustration when perceiving a loss of personal freedoms. These findings highlight the role of anger and frustration in fueling resistance to vaccine mandates and health policies.

Some items related to coping and response to fear showed significant differences, such as “when I am afraid, I think about possible solutions”. Anti-vaccine individuals might focus on alternative solutions rather than accepting mainstream health interventions. And “when I am angry, I feel like protesting” describes how anti-vaccine individuals convene to protest or take action in response to their anger, reinforcing oppositional behavior. These insights suggest that their anger is often channeled into active resistance, such as protests or public opposition.

Based on the statistically significant findings from the ANOVA table, anti-vaccine individuals can be characterized by the following psychological traits:Strong Opposition to Authority: They highly value autonomy and resist perceived coercion from health authorities. Anti-vaccine individuals report significantly higher agreement with autonomy-related statements, reflecting strong resistance to external control.Low Trust in Institutions: There is a clear skepticism toward official information and a preference for peer-driven knowledge. Lower trust in official institutions and reliance on alternative sources highlight skepticism toward public health authorities.Heightened Fear and Anxiety: Concerns about bodily integrity, loss of control, and societal pressures. As stated in the article by [33] ([33]), the symptomatic higher concern for anguish to be invaded supports the psychodynamic hypothesis that anti-vaccine people are deeply disturbed by things and ideas thrown into their bodies and minds.Emotional Reactivity (Anger and Protest): A tendency to react with frustration and protest when personal freedoms are perceived to be restricted. Anti-vaccine individuals exhibit higher levels of fear, anxiety, and anger, indicating emotional responses related to perceived threats to autonomy and personal freedoms.Social Polarization: Preference for like-minded communities that reinforce their beliefs.

### 4.2. Body Image and Schema Test Results

Table 3 reports items highly significantly different (*p* < 0.05) between pro-vaccine and anti-vaccine sub-samples in filling Body Image and Schema Test. The Body Image and Schema Test (BIST) is an assessment tool designed to evaluate individuals’ perceptions, attitudes, and representations of their body image and body schema. The test differentiates between body image, the subjective perception and emotional attitudes individuals hold about their physical appearance, which can be influenced by psychological, social, and cultural factors, and the body schema, the unconscious, sensorimotor representation of the body used for movement and spatial orientation, rooted in neurological and proprioceptive processes. The BIST focuses on identity’s personal and embodied dimensions ([19]), revealing how individuals negotiate body perceptions within cultural contexts. Grounded in psychodynamic theoretical frameworks, the BIST draws upon psychological constructs such as drive energy, self-perception, relational and bonding dynamics, body image, and embodiment. The internal organization of the BIST includes multiple subscales (seven factors) and items (25) that have been developed to capture both explicit and implicit body-related processes. The BIST consists of 25 strings of four pictures each. The authors have selected pictures as body-dynamic symbols. The BIST test provides insights into how individuals perceive their bodies, regulate their emotions, and form relationships with others. It helps in understanding the balance between internal energy, self-protection, and social belonging, offering valuable information for personal development and psychological well-being.

**General Psychological Tension (sum of all BIST items, line 1 of Table 3)** represents the overall score that summarizes all the aspects measured by the test. It provides a general picture of how individuals relate to their body image and emotional responses. Anti-vaccine individuals show a significantly higher overall BIST score, suggesting stronger emotional and cognitive responses related to body image and self-schema. This indicates a heightened sensitivity to body-related concerns.

**The Internal Body Model super-factor (line 2 of Table 3)** measures how people perceive their body’s internal sensations and energy. It reflects how aware individuals are of their body’s inner processes, such as feelings of vitality, physical health, and internal emotional states. Anti-vaccine individuals have a higher score, indicating a more intense perception of their body’s internal states. This may be linked to increased body awareness and a heightened concern about physical health. This could contribute to their search for bodily integrity. Three sub-factors constitute the Internal Body Model factor, and all three have significant statistical differences between anti-vaccine and pro-vaccine individuals.

**Anti-vaccine individuals display significantly higher levels of Drive Inner Energy (line 2 of Table 3)**, which refers to a person’s internal motivation and energy levels. It indicates how much drive an individual has to engage with life, pursue goals, and take action based on their inner needs and impulses, indicating a greater internal drive to act on their beliefs and emotions. ANOVA results suggest that anti-vaccine subjects are more likely to respond emotionally and assertively to perceived threats or challenges, including public health interventions. Their increased internal energy and drive possibly reflect heightened emotional activation and a stronger inclination to act upon internal beliefs and fears related to health interventions.

**The sub-factor Libidinal Extroverted Energy (line 4 of Table 3)** examines how people express their emotional and physical energy outwardly. It is associated with passion, social interactions, and the desire to connect with the world through actions and relationships. This factor assesses outwardly directed emotional energy. Anti-vaccine participants show greater emotional expressiveness, potentially reflecting assertiveness or even aggression toward authority. Anti-vaccine individuals exhibit greater Libidinal Extroverted Energy related to externalized emotional expressions and assertiveness. The pro-vaccine group, with lower scores, may rely more on internal reflection rather than emotional reactivity. This emotional intensity may explain their vocal and active opposition to public health mandates.

**The sub-factor Bonds Relational Energy (line 5 of Table 3)** measures how individuals experience and value emotional connections with others. It looks at how much they need relationships and their ability to establish meaningful connections in their social life. Anti-vaccine individuals exhibit higher Relational Energy, reflecting an intense need for interpersonal connections but potentially with ambivalent feelings towards close relationships. This may manifest as strong bonds within similar communities while rejecting mainstream social norms. For them, the need for relational bonding indicates that social belonging and shared beliefs within their community play a significant role in shaping their attitudes toward

The *item Bonding Energy Belonging (line 5.a. of Table 3)* is part of the Bonds Relational Sub-Factor. This sole item reaches a quite significant *p*-value (0.04). This item focuses on how much individuals feel they belong to a group or community. It evaluates their sense of inclusion and the importance of being part of social networks. Anti-vaccine individuals show a stronger need for belonging within specific social groups, often aligning themselves with communities that reinforce their perspectives. This suggests the importance of in-group validation in shaping their vaccine beliefs. Anti-vaccine individuals score higher in Bonding Energy, suggesting a desire for belonging while simultaneously resisting mainstream norms. This could manifest in aligning with alternative communities that oppose vaccinations.

**The Body Image factor (line 6 of Table 3)** assesses how individuals perceive their own physical body structure in terms of a partial or global entity. It reflects their thoughts, feelings, and concerns about their physical self-image. Anti-vaccine individuals have a heightened sense of their entire body, near to self-body idealization. Higher scores in this factor suggest that anti-vaccine individuals may reflect heightened concern related to physical diminutions due to health risks.

**The Ego Skin factor (line 7 of Table 3)** is the core factor used to interpret the characteristics of anti-vaccine subjects. This factor measures how individuals perceive their personal boundaries and sense of self-protection. It relates to their ability to feel safe and secure within their personal space and identity. Anti-vaccine individuals show pronounced Ego Skin defenses, indicating heightened vulnerability and sensitivity to perceived external threats. Higher scores in the anti-vaccine group suggest increased sensitivity to perceived threats to their personal Ego Skin. This may translate into heightened resistance to any external interventions they perceive as invasive.

*The item Ego Protection (line 7.a. of Table 3)* is one of the three items that form the Ego Skin factor. This item examines how much individuals need to defend themselves emotionally and physically. It measures their tendency to protect themselves from perceived threats or intrusions. Anti-vaccine individuals exhibit stronger defensive mechanisms, reinforcing their resistance to external influence and their need to protect their autonomy and health choices. Anti-vaccine individuals have significantly higher Ego Protection scores, reflecting a heightened defensive stance towards perceived threats to their autonomy and beliefs. This psychological mechanism could contribute to their resistance to persuasion and external influence.

The second of the three statistically significant items is the item *Ego Willingness (line 7.b. of Table 3)*. This item evaluates a person’s intention in confronting new situations and challenges. It measures the individual’s intentionality toward experiences and ideas. Lower scores in the pro-vaccine group suggest a greater openness to external influence and health recommendations. In contrast, the anti-vaccine group shows a more rigid stance regarding changing their perspectives. Anti-vaccine individuals exhibit higher personal intentionality, which could hinder their adaptation to external suggestions. This could reflect an over-investment of self-belief cognitive credence that could make resistance to change.

In conclusion, the BIST analysis highlights that anti-vaccine individuals exhibit stronger emotional responses, heightened body awareness, and increased resistance to external influence. Their social connections, self-protection mechanisms, and resistance to change contribute to their firm stance against vaccinations, emphasizing the importance of addressing these psychological aspects in public health communication strategies.

## 5. Discussion

The statistically significant differences in the ANOVA analysis provide a detailed understanding of the psychological traits characterizing anti-vaccine individuals. This profile integrates emotional, cognitive, and behavioral components, illustrating how these traits influence opposition to vaccination.

The anti-vaccine group is also characterized by individualism and an anti-authority attitude. The pro-vaccine group shows emotional stability, trust in authority, and collective responsibility. These findings emphasize how deep psychological dynamics influence health-related decisions.

Based on these significant differences, anti-vaccine individuals can be characterized as follows:Defensive and Guarded: Their high psychological defenses make them resistant to external control and skeptical of authority. Their stronger Ego Skin defenses reflect a need to protect themselves from perceived external threats.Increased Emotional Tension: Greater psychological strain makes them more reactive to health directives. They experience intense emotions, particularly fear and anger, and are quick to express them.Somato-Psychic Vulnerability: Amplified Internal Body Model scores suggest heightened bodily awareness, contributing to fears of bodily intrusion.Impulsive and Action-Oriented: Heightened internal drives push them toward immediate, emotionally charged actions rather than reflective thinking. Elevated Inner Drive Energy reflects emotional impulsivity and assertiveness.High Emotional Expressiveness: Enhanced Libidinal Energy indicates more outward emotional reactions, possibly manifesting as anger or defiance.Socially Vocal: They often express their beliefs assertively and may actively seek out like-minded communities that reinforce their views.

In contrast, pro-vaccine individuals demonstrate:Lower Psychological Defensiveness: A calmer and more adaptive approach to external influences.Emotional Stability: Less psychological tension and better emotional regulation.Balanced Somato-Psychic Perception: A more integrated body image reduces anxiety about bodily interventions.Moderate Internal Drive: Less impulsivity and greater emotional control.Controlled Emotional Expression: Lower outward emotional reactivity promotes rational decision-making.

These significant differences provide a comprehensive understanding of the psychological mechanisms underlying vaccine attitudes. Anti-vaccine individuals are driven by defensiveness, emotional intensity, and a bodily need for integrity, maybe as a reaction to a subjective sense of self-vulnerability. Conversely, pro-vaccine individuals exhibit emotional regulation, openness, and social responsibility.

Anti-vaccine psychological traits and beliefs (see Table 2 and Table 3) can explain their rejection of health mandates; viewing vaccination campaigns as an overreach of government control, they may refuse to comply with health mandates or participate in protests. The spread of misinformation contributes to sharing emotionally provocative anti-vaccine content online without verifying sources to confirm pre-existing attitudes and beliefs. Health Autonomy Advocacy that prefers “natural immunity” or alternative health remedies over medical interventions can be correlated to their strong intentionality, which projects personal credence to the social reality. The seeking for social support takes form in anti-vaccine communities, reinforcing their beliefs and reducing exposure to counterarguments.

A notable limitation of this study concerns self-report measures, which can introduce biases such as social desirability or recall inaccuracies. Participants may provide responses they believe are expected or socially acceptable rather than those reflecting their attitudes and behaviors. This can particularly affect sensitive topics like vaccine hesitancy, where societal pressures or stigma may influence how individuals report their beliefs and intentions. Additionally, validating the Body Image and Schema Test (BIST) poses another challenge. Although the BIST is instrumental in assessing body schema and related psychological constructs, it is still undergoing validation. This ongoing process may affect the reliability and generalizability of the findings derived from its use. This study’s reliance on a tool that is not yet fully validated means that interpretations of the data should be approached cautiously, as the test may not comprehensively capture all relevant dimensions of body image and schema as intended.

## 6. Conclusions

Anti-vaccine individuals are psychologically distinguished by their strong defensive mechanisms, emotional intensity, bodily vulnerability, and assertive behaviors. Their opposition to vaccines is not merely a stance against a medical procedure. It is deeply rooted in psychological traits that prioritize personal autonomy, amplify fear responses, and reject external authority.

Authoritarian resistance and individualism, also evident in higher scores for body schema representation and emotional regulation, elevate anger and fear responses, suggesting heightened emotional vulnerability. As the Body Image and Schema Test register, Body Schema representation reflects underlying anxieties and somato-psychic conflicts between the search for integrity and close fear of losing it.

In conclusion, this study’s findings underscore the complex interplay between psychodynamic factors and vaccine hesitancy. By integrating the Body Image and Schema Test (BIST) with a cross-cultural analysis, this research confirms and expands upon Allport’s theories regarding the continuity of personality traits and their influence on behavior, such as vaccine hesitancy. The data suggest that body schema is crucial in mediating the relationship between individual psychological profiles and attitudes towards health interventions.

Moreover, this study highlights the importance of considering individual psychological predispositions and the broader socio-cultural environment when addressing public health challenges. The nuances revealed by our analysis provide key insights for designing more effective health communication strategies sensitive to populations’ diverse psychological makeups. These strategies could involve personalized approaches that acknowledge and respect individual body schema perceptions and the psychological need for autonomy.

Future research should continue to explore the underpinnings of vaccine hesitancy, mainly through longitudinal studies that can assess changes over time and the effects of interventions tailored to specific psychological profiles. The insights from such studies will be invaluable in crafting public health policies and campaigns that seek to increase vaccine uptake while respecting individual autonomy and psychological well-being.

## Figures and Tables

**Table 1 behavsci-15-00493-t001:** Sample.

		Male (45%)		Female (55%)		
		Pro-Vaccine (32%)	Anti-Vaccine (13%)	Pro-Vaccine (40%)	Anti-Vaccine (15%)	TOT
	Years	N.	N.	N.	N.	N.
Italy	18–32	51	20	124	43	238
	33–49	33	11	54	27	125
	50–75	68	16	102	24	210
France	18–32	26	13	14	5	58
	33–49	23	4	15	6	48
	50–75	43	8	61	20	132
United Kingdom	18–32	36	12	22	22	92
	33–49	21	8	20	10	59
	50–75	44	3	65	3	115
Germany	18–32	20	12	18	10	60
	33–49	13	9	15	9	46
	50–75	58	12	59	6	135
Poland	18–32	39	50	13	22	124
	33–49	18	20	14	14	66
	50–75	36	15	50	29	130
TOT		529	213	646	250	1638

**Table 2 behavsci-15-00493-t002:** Statistically significant differences in subjects’ declarations between pro-vaccine and anti-vaccine sub-samples (items where the pro-vaccine mean is higher than anti-vaccine are in italics).

			Pro-Vaccine		Anti-Vaccine	
(*Items Where the Pro-Vaccine Sample Results in a Higher Mean Are Written In Italics*)	F	Sig.	Mean	SD	Mean	SD
I believe that no person should be obliged by any form of authority	79.00	0.00	3.48	1.27	4.10	1.48
The values an individual believes in cannot be overridden by the obligations of authority	202.69	0.00	3.32	1.28	4.28	1.33
The needs of collectivity are crushing those of individuals	64.95	0.00	3.53	1.36	4.11	1.44
I do not consider it proper for duties to override the will of an individual	100.55	0.00	3.29	1.20	3.94	1.40
Only the individual can decide how to protect his health	269.94	0.00	3.20	1.30	4.33	1.39
I want to be the owner to determine the values to follow	113.93	0.00	3.84	1.25	4.54	1.30
*Trust national and local government institutions*	220.96	0.00	3.23	1.21	2.28	1.31
*Trust official information*	250.12	0.00	3.27	1.18	2.27	1.32
Fear that someone will make decisions about my skin	130.61	0.00	3.55	1.39	4.41	1.46
Fear that the majority of the population determines my behavior	51.36	0.00	3.36	1.31	3.88	1.55
*When I am afraid, I feel like sharing with someone*	5.73	0.02	3.27	1.36	3.10	1.44
When I am afraid, I feel like thinking about possible solutions	38.10	0.00	3.23	1.44	3.71	1.55
*When I am afraid, I feel like finding a way to express myself*	5.32	0.02	3.30	1.29	3.14	1.37
The anguish of feeling invaded	11.17	0.00	3.93	1.37	4.18	1.45
*The anguish of not being estimated*	8.37	0.00	3.61	1.33	3.40	1.36
Does it make me angry that politics decides about people’s lives?	139.27	0.00	3.32	1.37	4.22	1.58
When I get angry, I feel like talking to someone else	8.28	0.00	3.10	1.39	3.32	1.45
When I get angry, I feel like thinking about a solution	26.07	0.00	3.10	1.41	3.49	1.47
*When I get angry, I feel bad about everything*	9.62	0.00	3.01	1.37	2.77	1.49
I get angry when I cannot achieve my goals	11.66	0.00	3.36	1.36	3.61	1.44
I get angry when limits are placed on me	17.73	0.00	3.49	1.35	3.80	1.41
I get angry when someone takes away something I care about	26.63	0.00	3.59	1.40	3.98	1.45
I get angry when I am not listened to	24.46	0.00	3.27	1.40	3.64	1.48
*I get angry when I have to leave a pleasant experience*	11.94	0.00	3.40	1.43	3.14	1.44
I get angry when I am pushed to do something I do not want to	85.42	0.00	3.30	1.43	4.02	1.52

**Table 3 behavsci-15-00493-t003:** Statistically significant differences in subjects’ Body Image and Schema Test responses between pro-vaccine and anti-vaccine sub-samples (in bold, factors; in italics, items forming the factor; values are expressed in z-point).

			Pro-Vaccine		Anti-Vaccine	
	F	Sig.	Mean	SD	Mean	SD
1. All BIST Items	11.146	0.001	−0.009	1.036	0.17	1.011
**2. Super-Factor Internal Body**	9.828	0.002	0.018	1.034	0.185	0.996
**3. Sub-Factor Drive Inner Energy**	6.922	0.009	0.063	0.992	0.199	1
**4. Sub-Factor Libidinal Extroverted Energy**	5.157	0.023	−0.003	1.03	0.117	0.992
**5. Sub-Factor Bonds Relational Energy**	6.915	0.009	−0.026	1.052	0.115	0.981
*5.a. Item Bonding Energy Belonging*	4.228	0.04	0.007	1.016	0.114	0.981
**6. Factor Body Image**	6.05	0.014	0.041	1.024	0.172	1.019
**7. Factor Ego Skin**	9.619	0.002	−0.057	1.019	0.107	1.017
*7.a. Item Ego Protection*	5.68	0.017	0.002	1.046	0.129	0.99
*7.b. Item Ego Willing*	4.715	0.03	−0.045	1.01	0.068	0.994

## Data Availability

The data supporting this study’s findings are available in the Open Science Framework repository: https://osf.io/vum2h/?view_only=216f282682ae409882d4c0d1439f8305 (accessed on 1 February 2025).

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
