# Peer review of "Body Image, Autonomy, and Vaccine Hesitancy: A Psychodynamic Approach to Anti-Vaccine Individuals’ Resistance"

_behavsci, 2025, doi:10.3390/bs15040493_

Round 1

Reviewer 1 Report

Comments and Suggestions for Authors

I appreciate the opportunity to read this manuscript, which provides some interesting perspectives on vaccine hesitancy. I liked the study's multi-theoretical arguments and the authors' efforts to streamline these arguments for the reader. Nevertheless, I have some concerns that I hope the authors will be able to address. A significant one is the absence of a complete presentation of the BIST theoretical framework and a full description of the meaning of BIST Factors and Items that the authors state they have included in the OSF repository. They do not exist - the only file available is the dataset (or, at least, I could not find them). It is challenging to interpret the data and corroborate the authors' findings without the instruments. Another concern is the lack of information about the instrument validation and the process to ensure that the data collected in five different European countries (with different languages, cultures, and beliefs) can be aggregated and studied. There is no mention in the introduction or the literature review about the motivation for the cross-country study. Although the authors included segmentation based on age and gender in the sample description, none of these demographic variables were used in the statistical analysis. Multiple studies link psychological reactance with age and gender, but this study seems to ignore this.

Furthermore, education has been shown to play an important role in vaccine hesitancy, but it has not been considered in this study.

The statistical analysis can be improved. There is no information about the sample's homogeneity of variance, the distribution's normality, or the observations' independence. 

Both the results and the discussion section contain statements that are assumptions but have never been explored as part of the study (or, if they have been, the analysis is not included in this manuscript). For example (158-159): "A marked distrust in official information sources is evident, reinforcing their reliance on alternative sources of information." Where was the reliance on alternative sources of information queried in this study?

Some minor observations that are easily remedied: a) include the percentage of the population when presenting demographic data (line 33 - 9+ million Italians are not vaccinated). What does that mean? There is no information about vaccine hesitancy in the other countries where the study was conducted. How was the sample recruited? How was consent to participate obtained, and what ethical body supervised the study? How was the questionnaire administered?

I hope the authors will be able to address my concerns, as I found the idea behind the study intellectually rewarding and enjoyable.

Author Response

Dear Reviewer, Thank you very much for your suggestions, and I'd like to offer my pardon for all the mistakes I made in the article.

First, I am very sorry that I did not open the OSF link to visitors, where the Body Image and Schema Test is quite fully presented (I say this because the test is under validation, and the whole manuscript is 120 pages… ).

Qualtrics made the European country sample, and the questionnaire and BIST were presented in each original language.

Segmentation by age and instructions was not taken into account for “space” reasons and because five education levels will restrict the class sample subject numbers.

The stochastic subject’s recruitment procedure assured by Qualtrics guaranteed the sample's homogeneity of variance, the distribution's normality, or the observations' independence. 

You said: Both the results and the discussion section contain statements that are assumptions but have never been explored as part of the study (or, if they have been, the analysis is not included in this manuscript). For example (158-159): "A marked distrust in official information sources is evident, reinforcing their reliance on alternative sources of information."

This affirmation comes from the t-test statistically significant difference between Pro-vax and Anti-vax in these two statements:

Pro-Vaccine

Anti-Vaccine

(In italics are written items where Pro-vaccine sample results with a higher mean) 

F

Sig.

Mean

SD

Mean

SD

Trust national and local government institutions

220,96

0,00

3,23

1,21

2,28

1,31

Trust official information

250,12

0,00

3,27

1,18

2,27

1,32

Some minor observations that are easily remedied: a) include the percentage of the population when presenting demographic data (line 33 - 9+ million Italians are not vaccinated). What does that mean? There is no information about vaccine hesitancy in the other countries where the study was conducted.

I added the information you requested in the manuscript.

Consent to participate and ethical commission have been declared to the Behavioural Science Submit Manuscript Form.

Reviewer 2 Report

Comments and Suggestions for Authors

Thank you for allowing me to provide feedback on the manuscript “Body Image, Autonomy, and Vaccine Hesitancy: A Psychodynamic Approach to Anti-Vaccine Individuals Resistance, “ submitted for consideration to Behavioral Sciences.

This manuscript aims to explore the psychological and psychodynamic factors underlying vaccine hesitancy, focusing on body image and emotions. It examines cognitive, emotional, and behavioral distinctions between pro-vaccine and anti-vaccine individuals to better understand their profiles. The findings seek to enhance the theoretical framework, aiding in creating strategies to mitigate potentially flawed decision-making behaviors.

The manuscript has potential, particularly regarding its ability to suggest certain enduring psychological determinants of vaccine hesitancy (e.g., personality traits), which is, in my opinion, an interesting contribution to the literature on vaccination behavior. This stands out in an original way from existing research, which emphasizes the nature of informational content and how it is processed as an explanatory factor for behavior. I believe both are essential for a better understanding of vaccine hesitancy. I particularly appreciated the efforts made in presenting the results; it is evident that the authors invested time in delivering the findings clearly and concisely, which is commendable.

After thoroughly reading and reflecting on the authors’ approach, I believe this paper holds the potential to provide valuable insights into vaccine behavior. Nevertheless, a significant revision needs to be completed. I believe the manuscript requires a thorough review to ensure it is relevant to a broad audience in the behavioral sciences.

My comments align with my work on the subject and fall within my area of expertise. I have refrained from addressing elements that lie outside my knowledge. Below, I suggest elements that could guide the extensive revision needed, focusing particularly on the theoretical positioning of the study within the broader literature on vaccine hesitancy.

The manuscript's most significant limitation lies in its theoretical foundation. The introduction is too minimalist to serve as a proper overview of relevant research, offering little guidance for the reader through the literature surrounding the study. The brief bibliography seems partly outdated (a third of the references predate 1980).

I agree that understanding the psychological foundations of vaccination acceptance and hesitancy is essential for developing targeted interventions (lines 48-49). Thus, gaining deeper insight into both pro-vaccine and anti-vaccine individuals' social and deeply ingrained psychological characteristics appears highly relevant. To do so, the authors should substantially update the cited works, emphasizing how their study integrates with recent literature.

I suggest that the authors familiarize themselves with recent studies that address the determinants of vaccine behavior and the psychological underpinnings of vaccine hesitancy, which go well beyond personality traits or intellectual orientations (lines 37-38). I recommend that the authors examine recent works proposing cross-sectional approaches to studying factors that influence vaccine hesitancy: the impact of pre-existing attitudes (Béchard, Gramaccia, Gagnon et al., 2024), the nature of vaccine-related information (Dubé, Macdonald, Manca, Bettinger et al., 2022; Tagiabue, Galassi, Mariani et al., 2020; Lee, Sun, Jang, & Connelly, 2022), as well as the effect of social media (Puri, Coomes, Haghbayan et al., 2020) and the level of mistrust (Murphy, Vallières, Bentall et al., 2021).

While I acknowledge the significance and influence of the classic works cited, many seem misaligned with current research on vaccination behavior. The seminal works of Solomon Asch (1951), Brehm (1966), and Stanley Milgram (1963), while significant for their contributions, fail to fully capture the timely relevance of the topic under investigation. Moreover, suggesting that Freud (1926) demonstrated how individuals with high anxiety unconsciously use strategies to manage internal conflicts and external stressors is, in my view, outdated and only moderately relevant to this study. I share similar views regarding Schilder (1935), so I would refrain from referring to these two.

The authors’ choice to rely on rather dated works is difficult to understand, especially given the substantial body of recent literature on the role of unconscious emotion in shaping behavior (e.g., Winkielman & Berridge, 2004). When not used to provide contextual background, grounding hypotheses in century-old studies always risks a disconnect between the approaches and methods of that era and what modern techniques can help refine the evidence from that time.

The main hypothesis is derived from Allport's classic work: Anti-vaccine individuals exhibit an anti-authoritarian personality structure, are more individualistic, experience invasion anxieties, and display pronounced anger and fear (lines 51-54). The connection between this hypothesis and Allport's work needs to be tightened because right now the reader struggles to grasp the specific concepts from Allport that are being mobilized to construct such a general hypothesis. Moreover, the discussion of Allport's classic model is too limited to help the reader fully understand the study's contribution. I suggest emphasizing how the study builds upon Allport’s established theories.

Considering that the primary objective of the paper is to describe and interpret attitudes toward COVID-19 vaccination through Allport’s work, which focuses on enduring personal characteristics influencing behavior (lines 40-42), it seems particularly relevant for the authors to consult recent works that have highlighted certain enduring determinants of vaccine behavior (e.g., Béchard, Gramaccia, Gagnon et al., 2024; Brannon, Tagler, & Eagly, 2007; Nazli, Yigman, Sevindik et al., 2022). This would help position the study within the latest research and provide a more comprehensive understanding of the psychological factors shaping vaccine attitudes.

These studies are particularly relevant, as they help further support Allport’s central idea of the continuity of personality traits and the concept of functional autonomy. These concepts emphasize that vaccine hesitancy is an attitude shaped by prior experiences and reinforced by current motivational states (lines 65-66). Integrating these recent studies will provide a broader theoretical foundation for understanding the persistence and evolution of vaccine hesitancy.

A few additional observations:

  • Readers would appreciate a more detailed description of the methodology used.
  • In my opinion, it is crucial for the authors to emphasize the significant limitation of using self-report measures to capture emotions. A brief limitations section would be helpful in better contextualizing the study's scope and addressing this potential drawback.
  • Lines 88 to 90 require proper citation.
  • Lines 97-100 should be moved to a footnote.
  • The presentation of Table 2 (italic vs. regular text) is confusing and hard to interpret.
  • The conclusion is incomplete and, in its current form, adds little value to the article.

Author Response

Dear Reviewer, Thank you very much for your suggestions. I want to offer my pardon for all the mistakes I made in the article and for all your precious suggestions on literature that I can now take into account.

The manuscript's most significant limitation lies in its theoretical foundation. The introduction is too minimalist to serve as a proper overview of relevant research, offering little guidance for the reader through the literature surrounding the study. The brief bibliography seems partly outdated (a third of the references predate 1980).

As you asked, I integrated Introduction, Theoretical framework and References.

I agree that understanding the psychological foundations of vaccination acceptance and hesitancy is essential for developing targeted interventions (lines 48-49). Thus, gaining deeper insight into both pro-vaccine and anti-vaccine individuals' social and deeply ingrained psychological characteristics appears highly relevant. To do so, the authors should substantially update the cited works, emphasizing how their study integrates with recent literature.

Thanks to your suggestions, I could add contributions by Béchard et al. and classical Psycho-social works (Asch, Freud et al., as Winkielman). Also, Allport’s personality theory was only partially integrated because embodied psychology, the fundamental framework of this research, exited Allport’s assumptions on personal characteristics influencing behaviour.

Here my answers to your observations:

  • Readers would appreciate a more detailed description of the methodology used.

I expand it in the article text.

  • In my opinion, it is crucial for the authors to emphasize the significant limitation of using self-report measures to capture emotions. A brief limitations section would be helpful in better contextualizing the study's scope and addressing this potential drawback.

Done and written in the text.

  • Lines 88 to 90 require proper citation.

Done.

  • Lines 97-100 should be moved to a footnote.

The journal style will not appreciate a footnote, but I will suggest it to the editor.

  • The presentation of Table 2 (italic vs. regular text) is confusing and hard to interpret.

I’ve written in the table: “In italics are items where the Pro-vaccine sample results with a higher mean.”

  • The conclusion is incomplete and, in its current form, adds little value to the article.

Conclusion have been completed.

Round 2

Reviewer 1 Report

Comments and Suggestions for Authors

I appreciate the work conducted by the authors to address my observations to their original manuscript. I also want to state that the authors do not need to apologize and that I never considered any of my observations as being the authors' "mistakes." It is just part of the review process.  It is my consideration that they have adequately responded to my comments and suggestions in their revised manuscript, especially the added information about sampling, data collection, and access to the BIST on OSF. I have one minor suggestion, which should be easy to address: in the Discussion section, please include a paragraph on the limitations of the study: the BIST being in the process of validation, and the self-reported nature of the instrument. The authors include these in the Methods section, but I think it would be better to move them to the Discussion section. Nevertheless, it is just a suggestion about structure, not content. I congratulate the authors for a very interesting study and an innovative approach to exploring vaccine hesitancy.

Author Response

Dear reviewer, Thank you for your comments on the article I proposed. Especially considering that the primary test used, the BIST, is still being validated, among other things, for a difficult object of study, such as bodily experience. I have specified the limits of the research in the Discussion.

Reviewer 2 Report

Comments and Suggestions for Authors

I believe the manuscript is of better quality than the first version. The introductory section dedicated to the literature review now better reflects the dynamism of recent years in research on vaccine behavior. I appreciate the addition of vaccination statistics by country in the introduction, which supports the author's initial argument, as well as the introductory addition in lines 57 to 66, which anchors the introduction within recent studies on vaccine behavior. I also appreciate the author's effort to clarify the methodology, measurement instruments used, and the limitations related to their use. Finally, I acknowledge the attempt to turn the conclusion into a proper recapitulatory section.

However, there are still some blind spots that I would like to share with the author before the manuscript can be accepted for publication. I consider these revisions mandatory.

I remain unconvinced of the relevance of referring to Freud’s and Schilder’s "insights" on how individuals manage inner conflicts. I still believe that citing Vaillant’s (1992) work is more than sufficient in this regard and that using century-old references to frame the theoretical framework does not do justice to the scope of the study (see lines 131 to 136). This point seems even more relevant now that the author has better clarified the use of the BIST, making the two aforementioned authors appear outdated and redundant.

Some suggestions for clarification:

  • Line 196: I would delete “all due to Qualtric's precise recruitment.”
  • I suggest the author carefully review the references cited in the manuscript. Some references appear to be non-existent, while others may contain errors. Specifically, I have doubts regarding the following references, which I was personally unable to find in bibliographic databases:
    • Béchard, D., Gramaccia, G., & Gagnon, S. (2024b). Implications of pre-existing psychological attitudes on public health interventions: The case of vaccine hesitancy. Journal of Health Psychology, 29(1), 112-123.
    • Tagliabue, F., Galassi, L., Mariani, P., et al. (2020). The role of individual differences, demographic, and social factors in predicting the acceptance of COVID-19 vaccination: An Italian survey. Vaccine, 38(49), 7788-7796.

Be cautious, as this could lead readers to suspect that an AI-based referencing tool was used for the literature review. As researchers readily acknowledge, such tools frequently generate fabricated references to non-existent studies.

In this regard, I suggest the author remove lines 106 to 112.

Author Response

Dear reviewer, Thank you for all your suggestions, which have helped improve the writing and correct an error in the bibliographic citation that I attribute to my slightly aged EndNote. For example, the citation of the authors Tagliabue, etc., was correct only in the year of publication. I have deleted and corrected where you suggested. I accepted your suggestion to refer to more recent psychological literature, deleting the references to Freud and Schilder.